# OSOA: One-Shot Online Adaptation of Deep Generative Models for Lossless Compression

**Chen Zhang**[†]    **Shifeng Zhang**[†]    **Fabio M. Carlucci**[*]    **Zhenguo Li**[†]

[†]Huawei Noah's Ark Lab
{chenzhang10, zhangshifeng4, li.zhenguo}@huawei.com

## Abstract

Explicit deep generative models (DGMs), e.g., VAEs and Normalizing Flows, have shown to offer an effective data modelling alternative for lossless compression. However, DGMs themselves normally require large storage space and thus contaminate the advantage brought by accurate data density estimation. To eliminate the requirement of saving separate models for different target datasets, we propose a novel setting that starts from a pretrained deep generative model and compresses the data batches while adapting the model with a dynamical system for only one epoch. We formalise this setting as that of One-Shot Online Adaptation (OSOA) of DGMs for lossless compression and propose a vanilla algorithm under this setting. Experimental results show that vanilla OSOA can save significant time versus training bespoke models and space versus using one model for all targets. With the same adaptation step number or adaptation time, it is shown vanilla OSOA can exhibit better space efficiency, e.g., $47\%$ less space, than fine-tuning the pretrained model and saving the fine-tuned model. Moreover, we showcase the potential of OSOA and motivate more sophisticated OSOA algorithms by showing further space or time efficiency with multiple updates per batch and early stopping.

## 1   Introduction

Lossless compression has always been an important part in the information and communications technology industry and has become more and more essential with the data explosion of the Big Data era [10, 39, 40, 8]. Recently, a large number of machine learning approaches, predominantly deep generative models (DGMs), have been proposed for lossless compression [34, 48, 19, 17, 51]. As indicated by Shannon's source coding theorem, the compression limit is determined by the entropy of the data's ground truth distribution [30]; any mismatch between an approximate distribution for coding and the unknown ground truth distribution will cause extra coding cost. It is the powerful approximation ability of DGMs that enables more precise estimation of the ground truth distribution [23, 41, 24, 29] and enables DGMs based compression algorithms to outperform the conventional counter-parties in terms of the compression ratio.

However, in real production scenarios, both space and time efficiencies are desirable. While more space efficiency is provided by higher compression ratio with DGMs based lossless compression algorithms, training of the probabilistic models is time consuming. Training a separate model for each different dataset to compress would scale the expensive time cost linearly. Moreover, to decode the dataset, one has to store the probabilistic model together with the codeword of the data, which requires additional space for local decoding and additional transmission cost for remote decoding. Apart from one model per dataset, another extreme on the spectrum is using one probabilistic model

---

[*]Work done during employment at Huawei Technologies R&D UK

35th Conference on Neural Information Processing Systems (NeurIPS 2021).

for many different datasets. Since each dataset can follow a very different distribution, using one model for all the datasets could lead to high inefficiency in terms of compression ratio.

To solve the above dilemma, we propose the setting of One-Shot Online Adaptation (OSOA) of deep generative models for lossless compression. A DGM based lossless compression algorithm under the OSOA setting (referred to as OSOA algorithm) includes one pretrained deep generative model (referred to as base model) and one deterministic dynamical system. The base model should be able to capture essential structures of a class of datasets to serve as a good starting point of the adaptation process. The dynamical system is the mechanism that can update the current distribution to a new one based on the current batch at each step in the OSOA process. At each OSOA step, one first uses the current deep generative model to evaluate the probabilistic mass function values to compress the current data batch before using the same batch to update the model with the dynamical system, which gives the name of online adaptation. The online adaptation is only conducted for one epoch and a distribution sequence will be unrolled during this process. Moreover, the dynamical system should be designed to optimise the objective functional associated with the deep generative model so that the sequence could provide better approximations on the unrolling procedure. Under this new setting, we can start from a proper base model and benefit from improved compression ratio as the sequence being unrolled. As the sequence is fully determined by the base model and the dynamical system, this approach allows to not only eliminate the need of extensively training a new model on each new dataset, but also to bypass the associated model storage cost that would normally be required.

Apart from the novel setting formulation, we showcase with experiments on large datasets that a vanilla OSOA algorithm can significantly save time versus training bespoke models and save space versus using one model for all targets. Moreover, with the same adaptation step number or adaptation time, vanilla OSOA can exhibit better space efficiency, e.g., $47\%$ less space, than fine-tuning the pretrained model and saving the fine-tuned model. We also show that further trade-off between time and space cost is available with varies strategies, e.g., multiple updates per batch and early stopping, and motivate more sophisticated algorithms with preliminary explorations. Further, we use sample images from OSOA and baselines to justify adaptation based compression algorithms and to discuss societal considerations for DGMs based lossless compression.

The paper is organised as follows. In Sec. 2, we review background knowledge regarding deep generative models for lossless compression. In Sec. 3, we formulate the setting of OSOA, discuss associated codec designing principles and propose a vanilla OSOA algorithm. In Sec. 4, we validate the setting by comparison with baselines, explore the strategies for further trade-off and discuss societal considerations. In Sec. 5, we discuss some related work. In Sec. 6, we summarise the paper, offer insights into our findings and point out promising future work directions.

## 2  Preliminary

**Entropy Coding** The core component underlying any lossless compression algorithm is the entropy coder, which converts between the data symbols and codewords. Popular entropy coders include Huffman coding [21], arithmetic coding (AC) [54] and asymmetric numeral systems (ANS) [9]. Huffman coding mainly involves binary trees operations, which is usually more time efficient than AC and ANS but can admit suboptimal codelength for probabilities not being power of $1/2$ [30]. As a result, popular codecs adopted by DGM based algorithms are normally based on AC or ANS, e.g., [34, 19, 35]. The main difference between AC and ANS is that the former encodes and decodes in a first-in-first-out (FIFO) style while the latter performs in a first-in-last-out (FILO) style [48]. All entropy coders require the knowledge of the data symbol alphabet and the associated distribution, in order to conduct compression and decompression. It is worth noting that the distribution used for entropy coding should be discrete and the average codelength is expected to be the entropy of the distribution used for coding. Recall that for a discrete distribution with the probabilistic mass function $p(x)$, the (Shannon) entropy is defined as $\mathbf{E}_{x \sim p}[-\log p(x)]$. Moreover, the optimal average codelength, given by the entropy of the ground truth distribution, is achieved only if the ground truth distribution is used for coding as per Shannon's source coding theorem [30]. Specifically, compressing data from distribution $p(x)$ with distribution $q(x)$ will lead to the expected extra codelength at $\mathrm{KL}(p(x)||q(x))$.

**Coding with Explicit Deep Generative Models** Since the entropy coders require probability values, implicit DGMs, e.g., generative adversarial networks (GANs) [11], cannot be readily used for lossless

compression. For explicit DGMs, the learning principle is either maximum likelihood estimation (MLE) [41, 24, 16]:

$$\hat{\theta} = \arg\min_{\theta \in \Theta} \mathbf{E}_{x \sim p^*(x)}[-\log p_\theta(x)] \tag{1}$$

or maximising the evidence lower bound (ELBO) of the logarithm evidence $\log p_\theta(x)$ [23, 49]:

$$\hat{\theta}, \hat{\phi} = \arg\min_{\theta \in \Theta, \phi \in \Phi} \mathbf{E}_{x \sim p^*(x)}[-\mathcal{L}(\theta, \phi, x)], \tag{2}$$

where $\mathcal{L}(\theta, \phi, x) = \mathbf{E}_{q_\phi(z|x)}[-\log q_\phi(z|x) + \log p_\theta(x|z)p_\theta(z)]$. Note that $x$ is a discrete random variable, so $p^*(x)$ refers to the probabilistic mass function of the ground truth data distribution.

When learning with MLE, one can either directly stipulate a discrete model or stipulate a continuous model with the dequantization technique [50, 45]. If one stipulates a discrete model, where $p_\theta(x)$ denotes the probabilistic mass function of the model, the objective function in Eq. 1 delegates the expected codelength when using $p_\theta(x)$ for entropy coding. If one stipulates a continuous model, one normally dequantizes the discrete data $x$ with a continuous noise $u$ uniformly distributed on the unit hypercube, and gets the dequantized variable $y = x + u$ with distribution $p^*(y)$. After fitting the probabilistic density function $p_\theta(y)$ of the model to the the probability density $p^*(y)$ of the dequantized distribution, one needs to discretise $p_\theta(y)$ and use the discretised distribution for entropy coding. In this case, the MLE objective $\mathbf{E}_{y \sim p^*(y)}[-\log p_\theta(y)]$ delegates an upper bound of the expected codelength [45]. For how to design sophisticated dequantization schemes or how to design discrete deep generative models, we refer the readers to [16, 18, 19, 51, 17, 55].

When learning with ELBO, one usually stipulates discrete models for $x$, i.e., $p_\theta(x|z)$ is a probabilistic mass function, and continuous models for $z$, i.e., $q_\phi(z|x)$ and $p_\theta(z)$ are probabilistic density functions. Furthermore, one usually does not have direct access to the close form of $p_\theta(x)$ but the generative components $p_\theta(x|z)$ and $p_\theta(z)$. To use $p_\theta(x|z)$ and $p_\theta(z)$ for efficient coding, a technique called bits-back [53, 15] is employed, which additionally involves the approximate posterior distribution $q_\phi(z|x)$ learned together with the generative components [48, 49, 26]. In these cases, the discretisation precision of the latent variable will not effect the average codelength, delegated by the objective in Eq 2, with consistent discretisation adopted for the latent distributions [48]. It is worth noting that, since bits-back works more compatibly with ANS, bits-back ANS [47] is usually the first choice for models learned with the ELBO functional.

## 3 One-shot online adaptation

In this section, we formulate the setting into three stages, i.e., 1) Pretraining, 2) OSOA Encoding and 3) OSOA Decoding, and summarise the OSOA Encoding / Decoding stages in Alg. 1. To unify the narrative of FIFO style and FILO style entropy coders, we aggregate the operations in Stage 2&3 into two modules, namely the dynamical system module $\mathcal{D}$ and the codec module $\mathcal{C}$. The deterministic dynamical system module $\mathcal{D}$ contains an optimiser, which can be a gradient based optimiser or a learning based meta optimiser, and associated parameters, to update a model $p_{\theta_{t-1}}(x)$ to $p_{\theta_t}(x)$ with a data batch $B_t$. The codec module $\mathcal{C}$ contains an entropy coder, a codeword variable and methods to modify the codeword with the entropy coder, cf. following discussions and Alg. 2.

---

**Algorithm 1** One Shot Online Adaptation: Encoding and Decoding

---

**OSOA Encoding**
1: **Inbuilt attributes:** $p_{\theta_0}(x), \mathcal{D}, \mathcal{C}$
2: **Input:** data $\{x_i\}_{i=1}^N$
3: Form data batches $\{B_t\}_{t=1}^T$ from data $\{x_i\}_{i=1}^N$
4: **for** $t = 1$ **to** $T$ **do**
5: $\quad \mathcal{C}.\texttt{encode\_or\_cache}(p_{\theta_{t-1}}(x), B_t);$
6: $\quad p_{\theta_t}(x) = \mathcal{D}(p_{\theta_{t-1}}(x), B_t);$
7: **end for**
8: **Output:** $c_f$ from $\mathcal{C}$.

**OSOA Decoding**
1: **Inbuilt attributes:** $p_{\theta_0}(x), \mathcal{D}, \mathcal{C}$
2: **Input:** code $c_f$
3: Initialise $\mathcal{C}$ with $c_f$
4: **for** $t = 1$ **to** $T$ **do**
5: $\quad B_t \leftarrow \mathcal{C}.\texttt{decode}(p_{\theta_{t-1}}(x));$
6: $\quad p_{\theta_t}(x) = \mathcal{D}(p_{\theta_{t-1}}(x), B_t);$
7: **end for**
8: **Output:** data $\{x_i\}_{i=1}^N$ from $\{B_t\}_{t=1}^T$

---

**Stage 1: Pretraining** A base model $p_{\theta_0}(x)$ is obtained by solving the problem defined in Eq. 1 or Eq. 2 on a pretraining dataset.

**Stage 2: OSOA Encoding** A sequence of distributions $\{p_{\theta_t}(x)\}_{t=1}^T$ will be unrolled through the adaptation procedure by the deterministic dynamical system $\mathcal{D}$ with batches $\{B_t\}_{t=1}^T$. Specifically, at each step $t$, one 1) inputs the model $p_{\theta_{t-1}}(x)$ and the data batch $B_t$ to the codec module and 2) performs the dynamical system $\mathcal{D}$ on $p_{\theta_{t-1}}(x)$ with $B_t$ and obtains the updated model $p_{\theta_t}(x)$. In 1), the method $\mathcal{C}.\texttt{encode\_or\_cache}$ determines whether to directly encode the current batch (FIFO style) or save the needed information, i.e., the variable values and associated probabilistic mass function (pmf) values, for asynchronous encoding (FILO style), cf. Alg. 2. At the end of OSOA Encoding, the final state of the codeword variable $c_f$ will be output from the codec module.

**Stage 3: OSOA Decoding** One uses the codeword $c_f$ from the OSOA Encoding stage to initialise the codeword variable in $\mathcal{C}$ and starts the iterative decoding process as follows. At each step $t$, one 1) uses the method $\mathcal{C}.\texttt{decode}$ to decode $B_t$ from the codeword variable with $p_{\theta_{t-1}}(x)$ and 2) updates the model with the decoded batch $B_t$ by the dynamical system $\mathcal{D}$. As a result, the same sequence of distributions $\{p_{\theta_t}(x)\}_{t=1}^T$ will be unrolled and all the data batches $\{B_t\}_{t=1}^T$ will also be decoded.

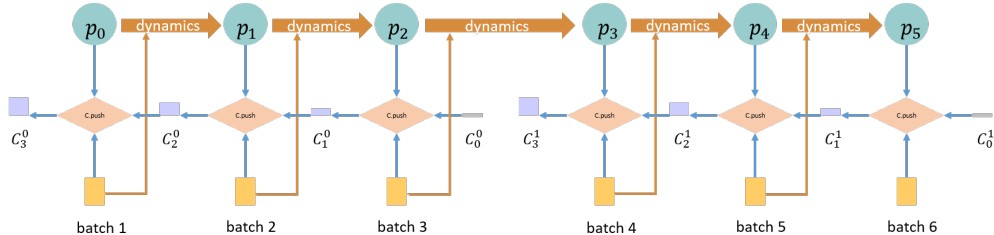

Figure 1: An illustration of OSOA Encoding with FILO style entropy coders, where $m = 3$ and $T = 6$ (bb-ANS as an example). C.push denotes the encoding operation of the ANS codec. We refer readers to Appendix A for complete demos of FIFO style and FILO style entropy coders.

**Methods in the codec module** To recover the exact distribution sequence $\{p_{\theta_t}(x)\}_{t=1}^T$ in the decoding stage, one has to decode the data batches in the order of $\{B_1, B_2, ..., B_T\}$. For DGMs based algorithms with an FIFO style coder, e.g., AC, one can consecutively encode the batch sequence $\{B_t\}_{t=1}^T$ in the original order and recover the correct order in the decoding stage. However, for FILO style coders, e.g., ANS, one has to temporarily save associated pmf values for encoding purpose in cache and encode them with the entropy coder in the reverse order. To reduce the size of cache memory, one can collect $m$ consecutive batches into one chunk and reversely compress the batches within the chunk, cf. Fig. 1. In this way, one effectively compresses different chunks into different code files. We refer readers to Appendix A for complete demos of OSOA with FIFO and FILO style coders. To distinguish the difference between the implementations of the $\texttt{encode\_or\_cache}$ method in Alg. 1 for coders of these two styles, we summarise the processes in Alg. 2. Note that $\texttt{coder.encode}$ in Alg. 2 and $\mathcal{C}.\texttt{decode}$ in Alg. 1 are respectively encoding and decoding functions already implemented with DGMs for lossless compression [34, 48, 49, 26]. Due to this flexibility, one can see that OSOA is generally applicable to DGMs based lossless compression algorithms.

---

**Algorithm 2** The $\texttt{encode\_or\_cache}$ method in OSOA Encoding

---

**FIFO coders**

1: **Inbuilt attributes:** $\texttt{coder}$
2: **Input:** $p_{\theta_{t-1}}(x), B_t$
3: $\texttt{coder.encode}(p_{\theta_{t-1}}(x), B_t)$;

**FILO coders**

1: **Inbuilt attributes:** $\texttt{coder, cache}, m$
2: **Input:** $p_{\theta_{t-1}}(x), B_t$

3: $\texttt{cache.append}((p_{\theta_{t-1}}(x), B_t))$;
4: **if** $\texttt{len(cache)} = m$ **or** $\texttt{last\_batch}$ **then**
5:   **for** $p(x), B$ in $\texttt{reversed(cache)}$ **do**
6:     $\texttt{coder.encode}(p(x), B)$;
7:   **end for**
8:   $\texttt{cache.empty()}$
9: **end if**

---

**Example: Vanilla OSOA** Following formulations of the proposed setting, we present the most straightforward strategy to conduct OSOA, referred to as vanilla OSOA. Specifically, the base model $p_{\theta_0}(x)$ is pretrained on the same data modality, e.g., HiLLoC [49] pretrained on CIFAR10 for image compression; and the deterministic dynamical system $\mathcal{D}$ is a gradient based optimiser, e.g., AdaMax [22]. **Stage 1.** Find or train base model $p_{\theta_0}(x)$ for certain data modality. **Stage 2.** Perform OSOA with base model from the last stage. The dataset to compress $\mathcal{S}$ should be first split to $T$

batches, i.e., $\mathcal{S} = \{B_t\}_{t=1}^T$. For $1 \le t \le T$, we first use $p_{\theta_{t-1}}(x)$ to compute pmf values to encode the $t$-th batch. Then we use the $t$-th batch with the specific optimiser and certain learning rate to update and get the new model $p_{\theta_t}(x)$. **Stage 3.** Perform OSOA with base model from Stage 1. The data splitting strategy is the *same* as Stage 2. For $1 \le t \le T$, we first use $p_{\theta_{t-1}}(x)$ to decode the $t$-th batch from the code. Then we use the decoded batch with the *same* $\mathcal{D}$ (e.g. optimiser, learning rate, etc.) as Stage 2 to update and get the new model $p_{\theta_t}(x)$. Note the entropy coding part is omitted for narrative simplicity, and we refer readers to Appendix A for more details.

## 4 Experiments with vanilla OSOA

### 4.1 Vanilla OSOA versus baselines

**Datasets** The datasets for base model pretraining are the renowned natural image datasets CI-FAR10 [28] and ImageNet32 [7], including images of size $32 \times 32$. We obtain three target datasets randomly sampled from the large image dataset Yahoo Flickr Creative Commons 100 Million (YFCC100m) [46] to test the compression performance. We name the three subsets from YFCC100m by SET32/64/128, where SET32 means the dataset of image height and width both being 32. To remove potential artifacts introduced by previous image compression, we first obtain SET256 by cropping the central $256 \times 256$ part in $2^{17}$ RGB images with size larger than $256 \times 256$ and down-sample the obtained dataset into our targets, as conducted in [34]. We use `Pillow` to downsample the SET256 into SET128, SET64 and SET32 with the high quality filter `Image.ANTIALIAS`. The $2^{17}$ RGB images are selected only with their shapes and no other discretionary criteria.

**Experiments configurations** Since an OSOA algorithm can be regarded as a general strategy to make a *static* deep generative model *dynamic*, it can be used for any DGMs based lossless compression frameworks. We adopt two representative DGMs based compression frameworks, HiLLoC [49] for hierarchical VAEs and IDF++ [51] for normalizing flows, to validate the OSOA setting. Specifically, for HiLLoC, we use the ResNet VAE (RVAE) model, which is proposed in [25] and adopted for compression in [49], with 24 layers and bi-directional connections for inference (a.k.a. top-down inference). Moreover, we showcase for a proof-of-concept purpose the IAF RVAE [25], with the same configuration as the RVAE model except for additional IAF layers in the inference model, making the approximate posterior more expressive. For IDF++, we use the same architecture proposed in [51], with 24 stacked flow layer blocks, where each block contains a permutation layer and a coupling layer. For each coupling layer in the flow model, a 12-layer Densenet [20] with 512 feature maps is used. For narrative simplicity, we use HiLLoC and IDF++ to refer to both the frameworks and the associated models. We use an Nvidia V100 32GB GPU for HiLLoC (and IAF RVAE) and an Nvidia V100 16 GB GPU for IDF++. Capped by the GPU memory limit, the batch size should be reduced as the image size grows. We quadruple the batch size as the image size decreases, i.e., batch size 256/64/16 in HiLLoC and batch size 48/12/3 in IDF++, for SET32/64/128 respectively.

Table 1: Theoretical bits per dimension (bpd) values of HiLLoC, IAF RVAE and IDF++ models training from scratch (ReTrain) on the three target datasets SET32/64/128.

| HiLLoC | | | IAF RVAE | | | IDF++ | | |
|---|---|---|---|---|---|---|---|---|
| SET32 | SET64 | SET128 | SET32 | SET64 | SET128 | SET32 | SET64 | SET128 |
| 2.668 | 2.504 | 2.521 | 2.608 | 2.292 | 2.392 | 2.361 | 2.443 | 2.422 |

**Baselines** We adopt three baselines, referred to as ReTrain, PreTrain and FineTune. For ReTrain, we train a separate model for each target dataset from scratch and use the final model to compress the target datasets. For PreTrain, we directly use the pretrained models to compress the target datasets. For FineTune, we fine tune the pretrained model on the target datasets and use the final model to compress the target datasets. Depending on how to set the number of fine-tuning epochs, we have three versions of FineTune. For FineTune v1, we fine tune the pretrained model for 2 epochs, as the whole OSOA Encoding & Decoding procedures involve 2 epochs of adaptations in total. For FineTune v2, we fine tune the pretrained model for 4 epochs for HiLLoC (and IAF RVAE) and 3 epochs for IDF++ to make the model updating time comparable with OSOA. This is because the dynamical system module in vanilla OSOA needs strict determinism [1] and it could cause extra time

Table 2: Theoretical bpd values between vanilla OSOA and baselines (PreTrain, FineTune v1, v2 and v3). The bpd values of baselines less than those of vanilla OSOA are shown in green, e.g., $0.132 = 3.505 - 3.373$. The effective bpd values by saving the models with FineTune baselines, defined as `#trainable_parameters` $\times$ `bits/parameter` $\times 1/$`#dataset_total_dims`, are shown in red. HiLLoC CIFAR10 denotes the CIFAR10 pretrained HiLLoC model.

| | SET | PreTrain | OSOA | FineTune v1 | FineTune v2 | FineTune v3 | Model |
|---|---|---|---|---|---|---|---|
| HiLLoC CIFAR10 | 32 | 3.830 | 3.505 | 3.373 (0.132) | 3.322 (0.183) | 3.215 (0.290) | 3.258 |
| | 64 | 3.561 | 3.090 | 2.954 (0.136) | 2.907 (0.183) | 2.800 (0.290) | 0.815 |
| | 128 | 3.268 | 2.682 | 2.564 (0.118) | 2.518 (0.164) | 2.412 (0.270) | 0.204 |
| HiLLoC ImgNet32 | 32 | 3.739 | 3.537 | 3.398 (0.139) | 3.329 (0.208) | 3.196 (0.341) | 3.258 |
| | 64 | 3.430 | 3.109 | 2.950 (0.159) | 2.893 (0.216) | 2.769 (0.340) | 0.815 |
| | 128 | 3.104 | 2.682 | 2.534 (0.148) | 2.482 (0.200) | 2.362 (0.320) | 0.204 |
| IAF RVAE CIFAR10 | 32 | 3.930 | 3.378 | 3.245 (0.133) | 3.185 (0.193) | 3.059 (0.319) | 3.963 |
| | 64 | 4.078 | 3.028 | 2.816 (0.212) | 2.751 (0.277) | 2.640 (0.388) | 0.991 |
| | 128 | 4.149 | 2.596 | 2.403 (0.193) | 2.354 (0.242) | 2.250 (0.346) | 0.248 |
| IAF RVAE ImgNet32 | 32 | 3.551 | 3.351 | 3.208 (0.143) | 3.146 (0.205) | 3.012 (0.339) | 3.963 |
| | 64 | 3.325 | 2.932 | 2.769 (0.163) | 2.707 (0.225) | 2.582 (0.350) | 0.991 |
| | 128 | 3.037 | 2.502 | 2.354 (0.148) | 2.300 (0.202) | 2.183 (0.319) | 0.248 |
| IDF++ CIFAR10 | 32 | 3.612 | 3.225 | 3.089 (0.136) | 3.045 (0.180) | 2.835 (0.390) | 4.517 |
| | 64 | 3.471 | 2.873 | 2.725 (0.148) | 2.687 (0.186) | 2.481 (0.392) | 1.166 |
| | 128 | 3.441 | 2.554 | 2.374 (0.180) | 2.335 (0.219) | - | 0.292 |
| IDF++ ImgNet32 | 32 | 3.662 | 3.338 | 3.160 (0.178) | 3.119 (0.219) | 2.921 (0.417) | 4.517 |
| | 64 | 3.439 | 2.949 | 2.768 (0.181) | 2.729 (0.220) | 2.528 (0.421) | 1.166 |
| | 128 | 3.401 | 2.623 | 2.448 (0.175) | 2.404 (0.219) | - | 0.292 |

cost in current deep learning frameworks. The time ratio we measured with/without the determinism is 1.98 (HiLLoC) in `TensorFlow 1.14` [4] with `tensorflow-determinism 0.3.0` [3] and 1.34 (IDF++) in `Pytorch 1.6` [2]. For FineTune v3, we fine tune the pretrained model for 20 epochs as an indicator of overfitting the pretrained model on target datasets. To take the additional model storage in FineTune baselines into consideration, we convert the model (in `float32`) storage size into effective bpd values and show them in the Model column in Table 2. Note that the bpd values shown under OSOA and baselines are of the codewords only, and the associated model bpd in the Model column need to be added to ReTrain and FineTune baselines for comparison of total storage cost. As bpd is the average number of bits needed to compress each dimension of the data, a lower value indicates a higher compression ratio. The time costs of OSOA and baselines are caused by model training, network inference (pmf evaluation) and entropy coding. We predominately use the model training time for comparison, as it dominates the time cost and the model and the entropy coder are controlled factors for comparisons.

Recent work [36, 6, 37] show that DGMs can assign high density values on data points not in the training datasets. And pretrained models can provide reasonable compression performance on unseen datasets [49, 55]. While PreTrain in Table 2 further confirms this point, the extra cost comparing to best possible ones can be significant, due to the distribution discrepancy between the training data and the target data. Nevertheless, PreTrain has the best time efficiency, since no extra training is needed.

Comparing to FineTune, OSOA does not require additional model storage for decompression. Fixing the batch order, the unrolled models in OSOA are the models in the first fine-tuning epoch, which tend to fit the data worse than the final one in the whole fine-tuning process (not necessarily true, cf. negative difference values in Fig 2 Right). Nevertheless, by comparing the decreased code space shown in green and the increased model space shown in red for FineTune v1 or FineTune v2 in Table 2, one can see that with the same optimisation budget or time budget, OSOA exhibits better total space efficiency than FineTune. For instance, with the CIFAR10 pretrained HiLLoC on SET32, OSOA takes $46.73\%$ less space than FineTune v2, with the same model training cost. To better understand the comparison between OSOA and FineTune v2 on SET128, we take the CIFAR10

pretrained HiLLoC and fix the order of images. If we only compress the first $2^{16}$, $2^{15}$ and $2^{14}$ images in SET128, the FineTune v2 admits bpd, in the code bpd + model bpd format, 2.564 + 0.408, 2.611 + 0.816 and 2.658 + 1.632. However, OSOA admits bpd, in the code bpd format since no model storage required, 2.734, 2.788 and 2.842, which are 0.238, 0.639 and 1.448 less than FineTune v2 respectively. Therefore, OSOA remains a clear net advantage than FineTune with the same time budget up to a reasonable large sample size even on large size images. From the comparison between OSOA and FineTune v3, one can see that the space advantage is still valid even if we conduct the FineTune for much longer time for SET32/64. FineTune v3 performs better on space efficiency on SET128, but it takes more than 4 times as much time as OSOA. Recent work on data lossy compression [52] introduces a model update coding term to the objective function and conducts lossless compression on the model update. However, this introduces an extra layer of complexity and time cost to determine suitable prior distributions for network parameters and to balance the data compression and model compression. Since OSOA does not require extra model saving, it exhibits huge advantage over the baselines on large-scale models or data of limited size. Note that OSOA achieves the optimal space & time cost balance with conducting model updates during data decoding. Therefore, it will be more suitable to scenarios of data backup or data cold storage than scenarios with real time data usage, e.g., video streaming.

At each OSOA step, the potential improvement of the model from the current batch can only benefit the compression of following batches. Moreover, the batch gradient can have a large variance and the stochastic gradient methods, e.g. SGD, Adam, etc., are not strictly descent algorithms. To investigate the online compression performance along the OSOA path, we showcase the results at the batch level in OSOA Encoding in Fig. 2 Middle and Right, with the CIFAR10 pretrained HiLLoC on SET32 and SET128. To make the bpd values comparable, we use the same batch order for OSOA, FineTune and PreTrain and showcase the differences between the bpd values of compressing each batch with OSOA and the baselines. Here we abuse the notation and use OSOA/FineTune to denote the vector of bpd values of each batch in OSOA/FineTune, with above fixed batch order. Then the 100th entry in the difference vector (OSOA - FineTune v1) is the extra cost of compressing the 100th batch with OSOA at the 100th OSOA Encoding step instead of FineTune v1 (with the fine-tuned model). Note that a negative difference value indicates that OSOA performs better than the corresponding baseline for that batch. From Fig. 2 Middle and Right, we can see that the advantage of compression with OSOA instead of PreTrain enlarges and the disadvantage (w.r.t. code space) of compression with OSOA instead of FineTune decreases, which validates the benefits brought by adaptation with previous batches to following batches in OSOA.

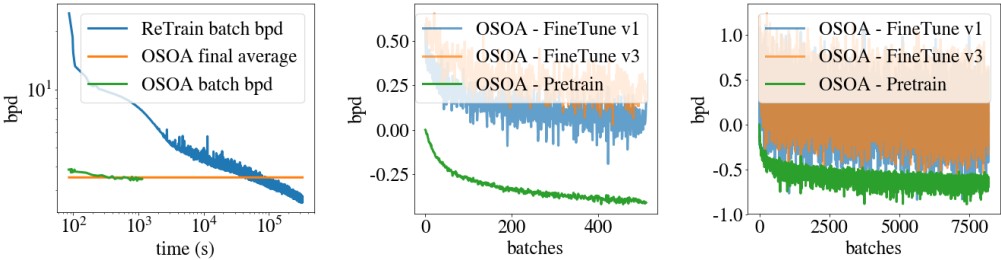

Figure 2: Left: Theoretical bpd values of each batch during training HiLLoC from scratch on SET32 (ReTrain HiLLoC) versus the batch bpd of OSOA, showing the significant time cost of training from scratch. OSOA final average is the bpd value after compressing all the batches with OSOA. Middle and Right: the differences between the theoretical bpd values of OSOA and the theoretical bpd values of the baselines of each batch in SET32 (Middle) and SET128 (Right) with the CIFAR10 pretrained HiLLoC. Negative values indicate the advantages of coding with OSOA instead of the corresponding baseline. The disadvantage of OSOA (versus FineTune) tends to decrease and the advantage of OSOA (versus PreTrain) tends to increase as the online adaptation being conducted.

Note that the theoretical bpd values reported in this paper are evaluated based on the discretised distributions of models for coding. Moreover, for HiLLoC, although the free-bits technique is used for the ResNet VAE model training, theoretical bpd evaluation does not involve free-bits. We showcase the real bpd values of HiLLoC and IDF++ on SET32 in Appendix B.

## 4.2 Vanilla OSOA variants to prioritise space or time efficiency

**OSOA with multiple updates per batch** Vanilla OSOA conducts one model update step per data batch with gradient based optimisers. However, it might not be sufficient for OSOA to exploit the adaptation potential from the base model, which is especially true for scenarios with small number of data batches. For instance, by comparing Fig. 2 Middle and Right, we can see that the performance improvement curves exhibit slower overall convergences on SET32 than SET128. To investigate the potential of OSOA in such scenarios, we follow the setting in Sec. 4.1 and conduct vanilla OSOA with multiple updates per batch. Note that each data batch is only used in one corresponding OSOA step and the multiple updates are conducted in that OSOA step, which will not change the one-shot nature and is different from multiple-epoch adaptation. In this section, we use SET32 with CIFAR10 pretrained HiLLoC and IDF++ and fix the batch size at 256 and 48 respectively. From Table 3, we can see that conducting multiple updates per batch can improve the performance for both HiLLoC and IDF++, and it can even further improve the bpd by around $0.4$ for HiLLoC. It is worth noting that in contrast to that overfitting does not happend for HiLLoC at 500 steps per batch, overfitting happens at 100 steps per batch for IDF++. Moreover, conducting multiple updates per batch scales the time cost linearly and it provides a modification to trade off time efficiency for space efficiency.

Table 3: The theoretical bpd of HiLLoC and IDF++ with OSOA of multiple optimisation updates per batch on test dataset SET32.

| updates/batch | 1 | 3 | 5 | 7 | 10 | 100 | 500 |
|---|---|---|---|---|---|---|---|
| HiLLoC (CIFAR10) | 3.505 | 3.420 | 3.382 | 3.357 | 3.332 | 3.184 | 3.109 |
| IDF++ (CIFAR10) | 3.225 | 3.137 | 3.096 | 3.080 | 3.046 | 3.105 | - |

**OSOA with early stopping** In vanilla OSOA, the number of adaptation steps is determined by the data size and the batch size. For very large datasets, additional bpd improvements from the latter OSOA steps may not be so significant, cf. Fig 2 Right. Due to the strict reproducibility of OSOA, if one sets a stopping criterion and uses the model at the stopping time for remaining data batches, the stopping criterion will be triggered at exactly the same step in OSOA Encoding and OSOA Decoding. Moreover, the time for model update, i.e., back propagation in the vanilla OSOA, will be saved for remaining batches. For instance, in the example of CIFAR10 pretrained HiLLoC for SET128, there are 8192 batches in total. If one stops the model adaptation at the 500th, 1000th, 2500th or 5000th batch, the final OSOA bpd will be 2.820, 2.774, 2.717 or 2.690 respectively, where one can accelerate the model adaptation efficiency by 8 times with the bpd cost less than $0.1$.

## 4.3 Discussion on DGMs for data lossless compression

Probabilistic modelling with DGMs for data generation encourages precise modelling of the target dataset for high fidelity generation. Different from generation purpose modelling, lossless compression purpose modelling only calls for high density values. The phenomena, DGMs assigning high density values for out-of-distribution (OOD) data [36, 6, 37] and plausible performance of pretrained models for OOD datasets lossless compression [49, 55], indicate that data generation purpose modelling and data compression purpose modelling are not strictly equivalent. And we refer the readers to [45] for an intuitive discussion on sample quality and log-likelihood values. It is discussed in [36] that DGMs weigh low-level statistics more than high-level semantics, and we show in above sections and Fig. 3 such inductive bias can be leveraged for lossless compression purpose. Derived from PreTrain, OSOA and FineTune admit samples with consistent semantics as PreTrain, fixing the random seed, cf. Fig. 3 (a), (b), (d) and (e). At the local feature level, e.g., sharpness and contrast, samples from the final models of OSOA and FineTune are more similar to samples from ReTrain, explaining improvements of OSOA and FineTune from PreTrain. While such local feature transfer largely originates from the small learning rate used in OSOA and FineTune, learning rate of too small or too large values will impede the performance, cf. Fig. 3 (f). Note that (b), (d) and (e) in Fig. 3 are hardly distinguishable, but they do differ in numerical values.

Another motivation of presenting samples in Fig. 3 is to discuss societal considerations to be addressed for future real applications of DGMs based lossless compression algorithms. Since trained DGMs model the distribution of the training set, information of the training data may be leaked by samples

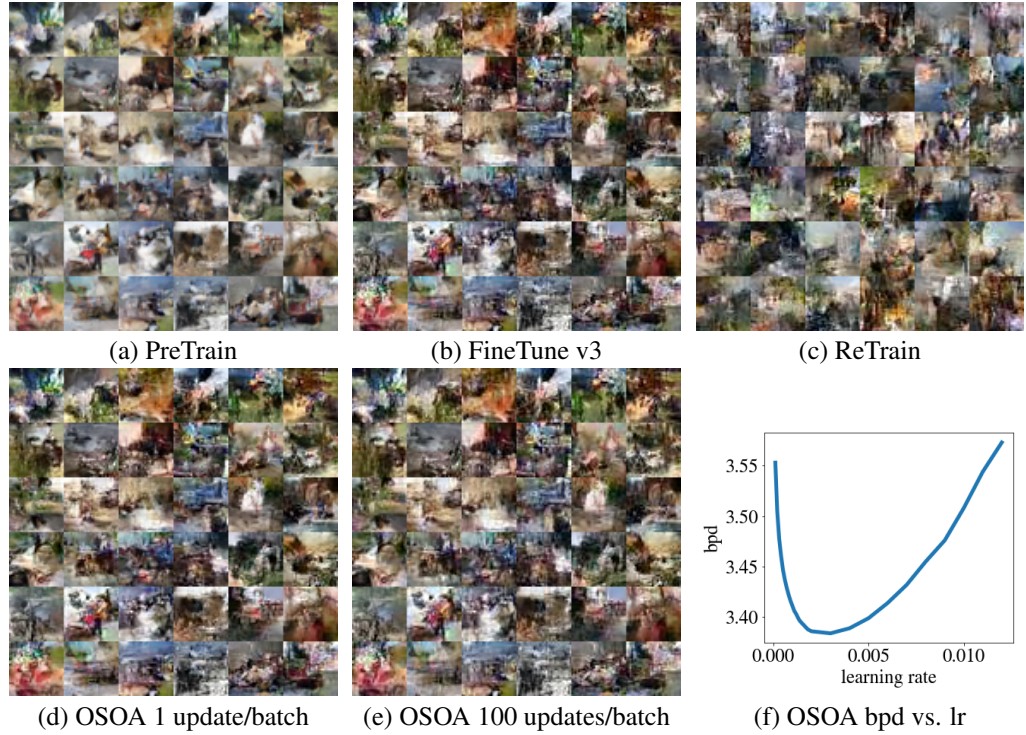

Figure 3: 36 images sampled from (a) PreTrain: CIFAR10 pretrained HiLLoC, (b) FineTune v3: fine-tuning CIFAR10 pretrained HiLLoC for 20 epochs on SET32, (c) ReTrain: HiLLoC trained on SET32 from scratch, (d) and (e) OSOA 1 and 100 updates/batch: the final checkpoint of vanilla OSOA with 1 and 100 update steps per batch from CIFAR10 pretrained HiLLoC, respectively. (f) The theoretical bpd values of vanilla OSOA with different learning rate values, with HiLLoC (CIFAR10).

from the trained models, e.g., the third row and fifth column in Fig. 3 (c) showing two people. Therefore, the specially trained models should be protected from privacy violation. From this perspective, pretrained models with as few adaptation steps as possible tend to have more privacy protection for service users. Nevertheless, information of the pretraining dataset can be retained in adapted models and influence the compression performance of target datasets. As a result, service providers should avoid using sensitive information in the pretraining dataset and avoid discrimination (different compression performances for data of different groups) caused by the pretraining dataset. As shown in this work, training on low resolution and non-sensitive images can offer good performance on images of different resolutions and semantics. Accordingly, it is promising to build societally friendly and effective DGMs base lossless compression algorithms with pretraining + adaptation style frameworks, e.g., OSOA, and careful incorporation of pretraining data.

## 5 Related work

OSOA de facto defines a paradigm to compress the distribution (model) sequence. The high level principle of it can trace back to a simple mathematical fact: a sequence can be uniquely determined by an initial point and a deterministic transition dynamics. In this section, we provide a non-exhaustive list of how this high level principle can be instantiated in compression related literature.

[43, 42] showcased the potential of neural networks as static context models for text compression and discussed the possibility of adaptive modelling with neural networks. [31] and DeepZip [44, 12] then instantiated the above principle for neural network based adaptive modelling with AC. This line of work focuses on the sequential modelling of entries in one data point and thus is intra-data adaptive modelling; we focus on DGMs of the data distribution and thus is inter-data adaptive modelling. Moreover, our work additionally bridges the initial probabilistic model with the adaptation process which enables the adaptation of a pretrained model and the benefit from pretrained models of datasets

sharing common structures. Another line of work is sequential data lossy compression like videos [52, 13]. They introduce encoder-decoder networks for compression, where the encoder network is adapted to sequential data during compression and the decoder network is used for decompression. However, the setting is quite different from OSOA. First, they are the lossy compression method and could only use VAEs for compression, while ours mainly deals with lossless ones and works on any explicit generative model. Second, they mainly deal with sequential data, while ours can compress any data batches, including the sequential ones.

The model storage cost for deep learning based data lossless compression is also considered in [5]. The work [5] formulates the problem in a supervised learning paradigm, where both the sender and receiver need the base model and an auxiliary dataset for the label transmission, whereas our work formulates in the unsupervised learning paradigm and only the base model is needed. Moreover, in order for the auxiliary dataset needed in [5] to effectively reduce the bitrate of the label (data to transfer), the label data needs high mutual information with the auxiliary dataset, which can form a stronger constraint.

The same principle also applies to pseudo-random generators, where a unique sample can be identified by the index of the sample generated from a shared pseudo-random source. [14] leveraged this fact and proposed an autoregressive style algorithm to compress the network parameter partition by partition with the sample indices of each partition and a predefined parameter prior. In the setting of lossy compression of Bayesian neural networks, [14] chooses one sample from the learned approximate posterior distribution to compress and thus deviates from the setting of data lossless compression where the data is given instead of chosen.

Prior to the developments in the deep learning literature, this high level principle has also been investigated for data compression. Adaptive dictionary methods, e.g., LZ77 [56], dynamically maintain a dictionary of substrings to gradually recover the optimal dictionary containing the repeated patterns in the sequence. Apart from dictionaries, the probability simplex can also be dynamically maintained to gradually converge to the ground truth probabilities or to capture the distribution shifts, e.g., context-adaptive binary arithmetic coding (CABAC) [38, 33]. Furthermore, for context mixture modelling methods, the mixture weights can be adapted during compression to favour the context models that make more accurate predictions, e.g., PAQ [32]. Due to the functional form of deep generative models, conducting OSOA of DGMs for lossless compression uniquely leverages the inductive bias of DGMs and could admit more flexible compression purpose modelling.

## 6 Conclusion and future work

In this work, we proposed the novel setting of One-Shot Online Adaptation (OSOA) of deep generative models for lossless compression and discussed codec designing principles for codecs of different styles. We showcased with a vanilla OSOA algorithm that OSOA can reduce the extensive model training time and model storage space for new datasets and achieve optimal trade-off between time and space cost versus alternative solutions. Our experimental exploration indicated that more sophisticated OSOA algorithms are promising and desirable. Finally, we discussed societal considerations for deep generative models based lossless compression algorithms.

The future work can be pursued in three folds. First, since the quality of the base model is very important for reducing the OSOA cost, it is highly desirable to study how to use deep generative models to capture the most of the common information shared among a general class of different datasets. Second, as shown in Sec. 4.2, there is still room to improve the space and time efficiency within one epoch of adaptation, it is thus of interest to study how more advanced dynamical systems, e.g., meta-learning methods or reinforcement learning methods, can further exploit the full potential of OSOA. Third, following the discussion in Sec. 4.3, it is interesting to investigate local feature transfer algorithms with pretrained models for fast lossless compression on unseen datasets.

## Acknowledgments and Disclosure of Funding

The authors thank the anonymous reviewers for their review and suggestions, which improved the quality of this manuscript. The authors thank Ning Kang and Mingtian Zhang for fruitful discussions.

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
