# Appendices to OSOA: One-Shot Online Adaptation of Deep Generative Models for Lossless Compression

## A  Appendix to Section 3

In this section, we adopt a bottom up explanation of OSOA and present two toy examples respectively for FIFO and FILO style entropy coders. For the bottom up explanation part, we start from principles and properties of common entropy coders, briefly summarise the ideas to synergise DGMs with the entropy coders proposed in recent work and arrive at how OSOA accommodates DGMs based lossless compression algorithms with different styles of entropy coders.

### A.1  Entropy coders

At the high level, the problem of lossless compression can be formulated as a problem finding a bijective map between the data domain and the code domain (the image of the data domain under such a map), such that the expected length of the variable in the code domain is less than the expected length of the variable in the data domain. And a general principle for achieving shorter expected length is to assign a shorter code to a more frequent symbol. Popular entropy coders, i.e., Huffman coding [3], arithmetic coding [10] and asymmetric numeral systems [1], define three different families of solutions to lossless compression under such principle.

**Toy Example 1** Here we use a simple example to explain the ideas and properties of above three entropy coders. Assume the variable $x$ has possible values $\{a_1, a_2, a_3, a_4, a_5\}$, with probabilities in Tab 1.

Table 1: The simplex of the Toy Example 1

| SYMBOL | $a_1$ | $a_2$ | $a_3$ | $a_4$ | $a_5$ |
|---|---|---|---|---|---|
| PROBABILITY | 0.32 | 0.08 | 0.16 | 0.02 | 0.42 |

**Huffman Coding** Huffman coding gives such a bijective map through a binary tree generation process. For a given probabilistic distribution, it starts from the two least probable symbols, $a_2$ and $a_4$, and forms a subtree $a_2 \bigvee a_4$, where the less probable symbol is the right leaf. The two symbols $a_2$ and $a_4$ are substituted by the subtree $a_2 \bigvee a_4$ with probability as sum of the two symbols, $0.10$. Then the same iteration is conducted until no more nodes can be added, cf. Fig 1. For a certain symbol, the code of it reads from the root to the leaf node, e.g., $0110$ for $a_2$. And the decoding process is to look up the codebook given by the tree. Note that Huffman coding is a prefix code, in the sense that no codeword is the prefix of another code. As a result, codewords of different symbols are isolated and a symbol sequence can be decode from the codeword sequence as the original order. For example, the code word for the symbol sequence $a_5 a_3 a_2$ is $10100110$.

**Arithmetic Coding** Arithmetic coders partition an interval into subintervals according to a simplex, associate one symbol with one subinterval and use a real number in that interval to represent the associated symbol. To encode a symbol sequence, arithmetic coders start from the unit interval $I_0 = [0, 1]$ and find the associated subinterval $I_1$ for the first symbol. Then $I_1$ will be partitioned again according to the simplex and a subinterval $I_2$ will be found for the second symbol. The iterative process is conducted until the end of the sequence and a real number in the last interval is used to

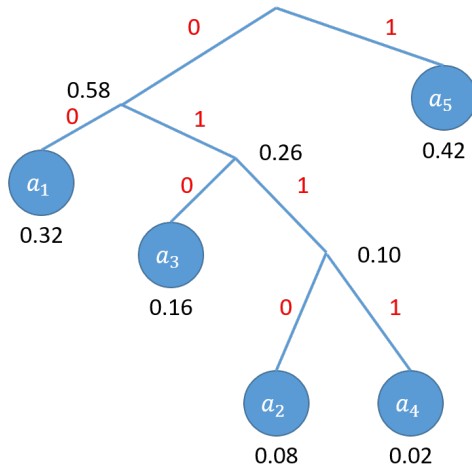

Figure 1: The Huffman tree for Toy Example 1.

represent the symbol sequence. To decode a sequence, one just need to examine which subinterval in $I_0$ the real number belongs to and decodes the associated symbol as the first one. Then one examines which subinterval in $I_1$ the real number belongs to and decodes the second symbol. To enable the iterative process to stop at the correct step, a termination symbol can be introduced. Without loss of generality, we assume $a_2$ is the termination symbol, where the codeword for $a_5 a_3 a_2$ by AC can be 0.77, cf. Fig. 2. Note that the full sequence $a_5 a_3 a_2$ is represented by a single codeword and one has to decode $a_5$ before being able to decode $a_3$. As a result, entropy coders of such fashion are referred to as of First-In-First-Out (FIFO) stytle, i.e., the queue style,.

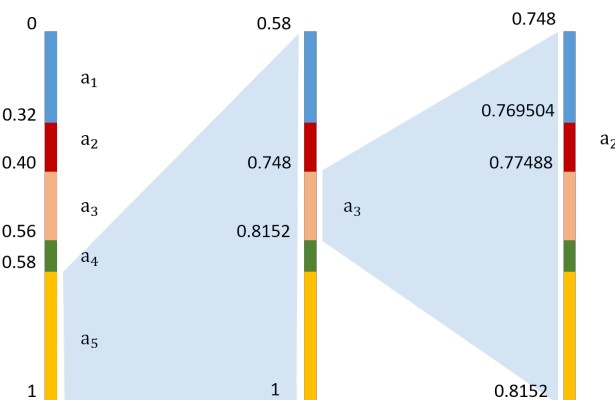

Figure 2: Arithmetic coding for Toy Example 1.

**Asymmetric Numeral Systems** Here we present rANS. rANS first approximates the probability of each symbol $a_i$ with a rational number $\frac{\ell_{a_i}}{m}$, where $m = \sum \ell_{a_i}$. The cumulative count is defined as $b_{a_i} = \sum_{j=1}^{i-1} \ell_{a_j}$. Furthermore, the inverse cumulative count function is defined as

$$b^{-1}(y) = \arg\min_{a_j}\{y < \sum_{i=1}^{j} \ell_{a_i}\}. \tag{1}$$

For a sequence $\{s_t\}_{t=1}^{T}$, the function to encode $s_t$ is

$$x_t = \lfloor \frac{x_{t-1}}{\ell_{s_t}} \rfloor * m + \texttt{mod}(x_{t-1}, \ell_{s_t}) + b_{s_t}. \tag{2}$$

And the function to decode $s_t$ is

$$s_t = b^{-1}(\texttt{mod}(x_t, m)), \tag{3}$$

$$x_{t-1} = \lfloor \frac{x_t}{m} \rfloor * \ell_{s_t} + \texttt{mod}(x_t, m) - b_{s_t}. \tag{4}$$

Note that $\{x_t\}_{t=0}^T$ is the codeword sequence generated during the encoding process and only $x_T$ is needed for decoding. The initial codeword state $x_0$ can be initialised as $0$.

Using the same Toy Example 1, we first compute the information as listed in Tab 2.

Table 2: Information needed of rANS for Toy Example 1

| SYMBOL | $a_1$ | $a_2$ | $a_3$ | $a_4$ | $a_5$ |
|---|---|---|---|---|---|
| PROBABILITY | 0.32 | 0.08 | 0.16 | 0.02 | 0.42 |
| $\ell_{a_i}$ | 32 | 8 | 16 | 2 | 42 |
| $b_{a_i}$ | 0 | 32 | 40 | 56 | 58 |

The encoding process for the sequence $a_5 a_3 a_2$ is as follows

1. $x_0 = 0$
2. $x_1 = \lfloor \frac{0}{42} \rfloor * 100 + \texttt{mod}(0, 42) + 58 = 58$
3. $x_2 = \lfloor \frac{58}{16} \rfloor * 100 + \texttt{mod}(58, 16) + 40 = 350$
4. $x_3 = \lfloor \frac{350}{8} \rfloor * 100 + \texttt{mod}(350, 8) + 32 = 4338$.

And the codeword for the sequence is $4338$.

The decoding process is as follows

1. $x_3 = 4338$
2. $s_3 = b^{-1}(\texttt{mod}(4338, 100)) = a_2$ and $x_2 = \lfloor \frac{4338}{100} \rfloor * 8 + \texttt{mod}(4338, 100) - 32 = 350$
3. $s_2 = b^{-1}(\texttt{mod}(350, 100)) = a_3$ and $x_1 = \lfloor \frac{350}{100} \rfloor * 16 + \texttt{mod}(350, 100) - 40 = 58$
4. $s_1 = b^{-1}(\texttt{mod}(58, 100)) = a_5$ and $x_0 = \lfloor \frac{58}{100} \rfloor * 42 + \texttt{mod}(58, 100) - 58 = 0$.

Different from Huffman coding and Arithmetic Coding, ANS decodes the symbols in the reverse order as the encoding one. As a result, the fashion of coding with ANS is referred to as of First-In-Last-Out (FILO) style, i.e., the stack style.

**Adaptive coding** We presented three coding examples respectively for each entropy coder with a fixed distribution of the symbols. All of the mentioned entropy coders allow for changes of the distributions during the coding process. For Huffman coding, it involves dynamic adjustments of the tree [5]. For AC, it involves different partitions at different steps [6]. For ANS, it involves different values of $\ell_{a_i}$'s and $b_{a_i}$'s [1]. While current DGMs based lossless compression algorithms only consider static coding, OSOA introduces and validates adaptive coding for DGMs based lossless compression algorithms.

## A.2 DGMs based lossless compression

The entropy coders reviewed in Sec A.1 are elementary functional units in a DGMs based lossless compression algorithm. As reviewed in Sec. 2, recent research focus on how to synergise above entropy coders with different types of DGMs, i.e., VAEs, Normalizing Flows, etc. The main reason is that for a data variable $\mathbf{x}$, normally we do not have direct access to the explicit form of $p(\mathbf{x})$ but instead some factorisation form of it in DGMs. Here we use a single latent VAE as an example to illustrate the components in a DGM based lossless compression algorithm, i.e., a deep generative model and an entropy coder and an algorithm to connect the model and the coder.

**Toy Example 2** In a single latent VAE, we have the observation variable $\mathbf{x}$ and the latent variable $\mathbf{z}$. The modelled distributions are prior $p(\mathbf{z})$, likelihood $p(\mathbf{x}|\mathbf{z})$ and the approximate posterior $q(\mathbf{z}|\mathbf{x})$.

For a chosen discretisation scheme, we denote the discrete distributions by $\bar{p}(\mathbf{z})$, $\bar{p}(\mathbf{x}|\mathbf{z})$ and $\bar{q}(\mathbf{z}|\mathbf{x})$. The entropy coder adopted here is rANS. The algorithm connects the VAE and rANS is called bits back ANS (bb-ANS) [7]. In bb-ANS, an auxiliary amount of initial bits are required, which is denoted by $c_0$.

To encode a given sample $\mathbf{x}_1$, one 1) decodes $\mathbf{z}_1$ from $c_0$ with $\bar{q}(\mathbf{z}|\mathbf{x})$ using rANS and gets the code $c_0^1$, 2) encodes $\mathbf{x}_1$ to $c_0^1$ with $\bar{p}(\mathbf{x}|\mathbf{z})$ using rANS and gets the code $c_0^2$ and 3) encodes $\mathbf{z}_1$ to $c_0^2$ with $\bar{p}(\mathbf{z})$ and gets $c_1$.

To decode $\mathbf{x}_1$ from $c_1$, since ANS is an FILO style coder, one has to reverse the above process and swap the operations of encoding and decoding. Specifically, one 1) decodes $\mathbf{z}_1$ from $c_1$ with $\bar{p}(\mathbf{z})$ using rANS and gets $c_0^2$, 2) decodes $\mathbf{x}_1$ from $c_0^2$ with $\bar{p}(\mathbf{x}|\mathbf{z})$ using rANS and gets $c_0^1$ and 3) encodes $\mathbf{z}_1$ to $c_0^1$ with $\bar{q}(\mathbf{z}|\mathbf{x})$ using rANS and gets $c_0$.

Due to the same reason of FILO coders, if one first encodes $\mathbf{x}_1$ with above algorithm and then encodes $\mathbf{x}_2$ with above algorithm, one can only first decode $\mathbf{x}_2$ and then decodes $\mathbf{x}_1$. Note that for FIFO coders, e.g., AC, one can only first decode $\mathbf{x}_1$ and then decode $\mathbf{x}_2$.

### A.3 OSOA

From Sec. A.2, one can see that a DGM based lossless compression framework adds one more layer, i.e., the deep generative model layer, to the entropy coder layer and synergise these two layers by an algorithm, e.g., bb-ANS in the above example. The proposed OSOA actually adds another layer, i.e., the model adaptation layer, to DGMs based lossless compression frameworks and synergise these two layers by different algorithms for FIFO style and FILO style entropy coders, which is adopted in a particular DGM based lossless compression framework. In this section, we present two simple examples respectively for DGMs based lossless compression frameworks with FIFO entropy coders and FILO entropy coders.

**Toy Example 3** We have three batches to compress, i.e., $B_1$, $B_2$ and $B_3$ and a DGM based lossless compression framework with a FIFO entropy coder. Moreover, we have a pretrained model $p_0$ for the DGM in the above framework.

The encoding include three steps as follows and illustrated in Fig. 3.

1. Use $p_0$ to compress $B_1$ with the inbuilt algorithm of the framework, then use $B_1$ to update $p_0$ and get $p_1$.
2. Use $p_1$ to compress $B_2$ with the inbuilt algorithm of the framework, then use $B_2$ to update $p_1$ and get $p_2$.
3. Use $p_2$ to compress $B_3$ with the inbuilt algorithm of the framework.

Note that since we do not have more batches, we do not need to use $B_3$ to update $p_2$.

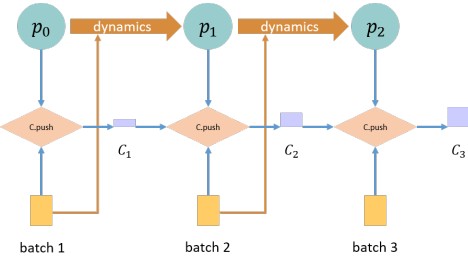

Figure 3: An illustration of OSOA encoding with arithmetic coding (AC) for Toy Example 3. C.push denotes the encoding operation of the AC codec.

The decoding include three steps as follows and illustrated in Fig. 4.

1. Use $p_0$ to decompress $B_1$ with the inbuilt algorithm of the framework, then use $B_1$ to update $p_0$ and get $p_1$.
2. Use $p_1$ to decompress $B_2$ with the inbuilt algorithm of the framework, then use $B_2$ to update $p_1$ and get $p_2$.

3. Use $p_2$ to decompress $B_3$ with the inbuilt algorithm of the framework.



Figure 4: An illustration of OSOA decoding with arithmetic coding (AC) for Toy Example 3. C.pop denotes the decoding operation of the AC codec.

**Toy Example 4** We have six batches to compress, i.e., $B_1$, $B_2$, $B_3$, $B_4$, $B_5$ and $B_6$ and a DGM based lossless compression framework with a FILO entropy coder. Moreover, we have a pretrained model $p_0$ for the DGM in the above framework. Here we divide the six batches into two chunks with consecutive three batches per chunk.

The encoding include six steps as follows and illustrated in Fig. 5.

1. Use $p_0$ to evaluate the pmf needed to compress $B_1$ with the inbuilt algorithm of the framework and add the pmf and batch to cache, then use $B_1$ to update $p_0$ and get $p_1$.

2. Use $p_1$ to evaluate the pmf needed to compress $B_2$ with the inbuilt algorithm of the framework and add the pmf and batch to cache, then use $B_2$ to update $p_1$ and get $p_2$.

3. Use $p_2$ to evaluate the pmf needed to compress $B_3$ with the inbuilt algorithm of the framework and add the pmf and batch to cache, then use $B_3$ to update $p_2$ and get $p_3$. Start an independent process to consecutively compress $B_3$, $B_2$ and $B_1$ with the information in the cache. Clear the cache for the first three batches.

4. Use $p_3$ to evaluate the pmf needed to compress $B_4$ with the inbuilt algorithm of the framework and add the pmf and batch to cache, then use $B_4$ to update $p_3$ and get $p_4$.

5. Use $p_4$ to evaluate the pmf needed to compress $B_5$ with the inbuilt algorithm of the framework and add the pmf and batch to cache, then use $B_5$ to update $p_4$ and get $p_5$.

6. Use $p_5$ to evaluate the pmf needed to compress $B_6$ with the inbuilt algorithm of the framework and add the pmf and batch to cache. Start an independent process to consecutively compress $B_6$, $B_5$ and $B_4$ with the information in the cache. Clear the cache for the last three batches.

Note that since we do not have more batches, we do not need to use $B_6$ to update $p_5$.

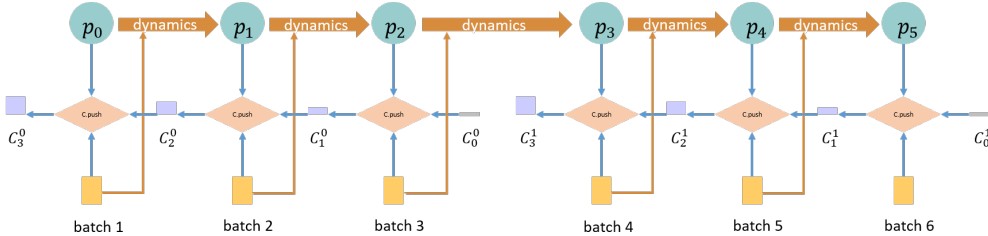

Figure 5: An illustration of OSOA encoding with bits back asymmetric numerical system (bb-ANS) for Toy Example 4. C.push denotes the encoding operation of the ANS codec.

The decoding include six steps as follows and illustrated in Fig. 6.

1. Use $p_0$ to decompress $B_1$ with the inbuilt algorithm of the framework, then use $B_1$ to update $p_0$ and get $p_1$.

2. Use $p_1$ to decompress $B_2$ with the inbuilt algorithm of the framework, then use $B_2$ to update $p_1$ and get $p_2$.

3. Use $p_2$ to decompress $B_3$ with the inbuilt algorithm of the framework, then use $B_3$ to update $p_2$ and get $p_3$.

4. Use $p_3$ to decompress $B_4$ with the inbuilt algorithm of the framework, then use $B_4$ to update $p_3$ and get $p_4$.

5. Use $p_4$ to decompress $B_5$ with the inbuilt algorithm of the framework, then use $B_5$ to update $p_4$ and get $p_5$.

6. Use $p_5$ to decompress $B_6$ with the inbuilt algorithm of the framework.

Note that since we do not have more batches, we do not need to use $B_6$ to update $p_5$.

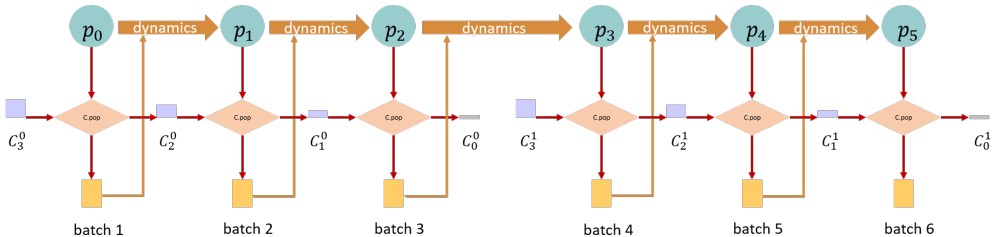

Figure 6: An illustration of OSOA decoding with bits back asymmetric numerical system (bb-ANS) for Toy Example 4. C.pop denotes the decoding operation of the ANS codec.

**Vanilla OSOA** In vanilla OSOA, the steps of using $B_i$ to update $p_{i-1}$ to $p_i$ will be using a gradient based optimiser to update $p_{i-1}$ to $p_i$ with the gradient evaluated on $B_i$, in both OSOA Encoding and OSOA Decoding.

# B  Appendix to Section 4

## B.1  Samples

Please find Fig 7 for visually larger images of those shown in Fig 3 in the main file.

## B.2  Existing assets

Existing assets used in this work include existing data and existing software.

**Data** Existing datasets used in this work, CIFAR10, ImageNet32 and YFCC100m, are public datasets freely used for research purpose. Both CIFAR10 and ImageNet32 are of the MIT license and YFCC100m is of Creative Commons licenses. SET32/64/128 are sampled and subsampled from images in YFCC100m as expained in Sec. 4.1. Since CIFAR10, ImageNet32 and SET32 are of very low resolution, images from these datasets can be regarded with almost no personally identifiable information. SET64 and SET128 are of higher resolutions than $32 \times 32$ and may include personal identifiable information, but we are not showing direct samples from them nor samples from models trained on them. Further, we discuss the importance of information protection with DGMs based lossless compression algorithms in Sec. 4.3 with samples of low resolution.

**Software** The open source software for deep learning and data preprocessing are included and cited in Sec. 4.1. Here we add more details on the code for deep generative models and associated lossless compression. The code for the RVAE model adopted in HiLLoC and the IAF RVAE model shown in this work are adapted from the official code release of the work [4] and the official code release of the work [8] both with the MIT license. The codec used with HiLLoC is the package Craystack released with [8] with the MIT license. The model and the codec used with IDF++ is released with IDF [2] in which slight modifications are performs to change IDF to IDF++. The codes are with MIT license.

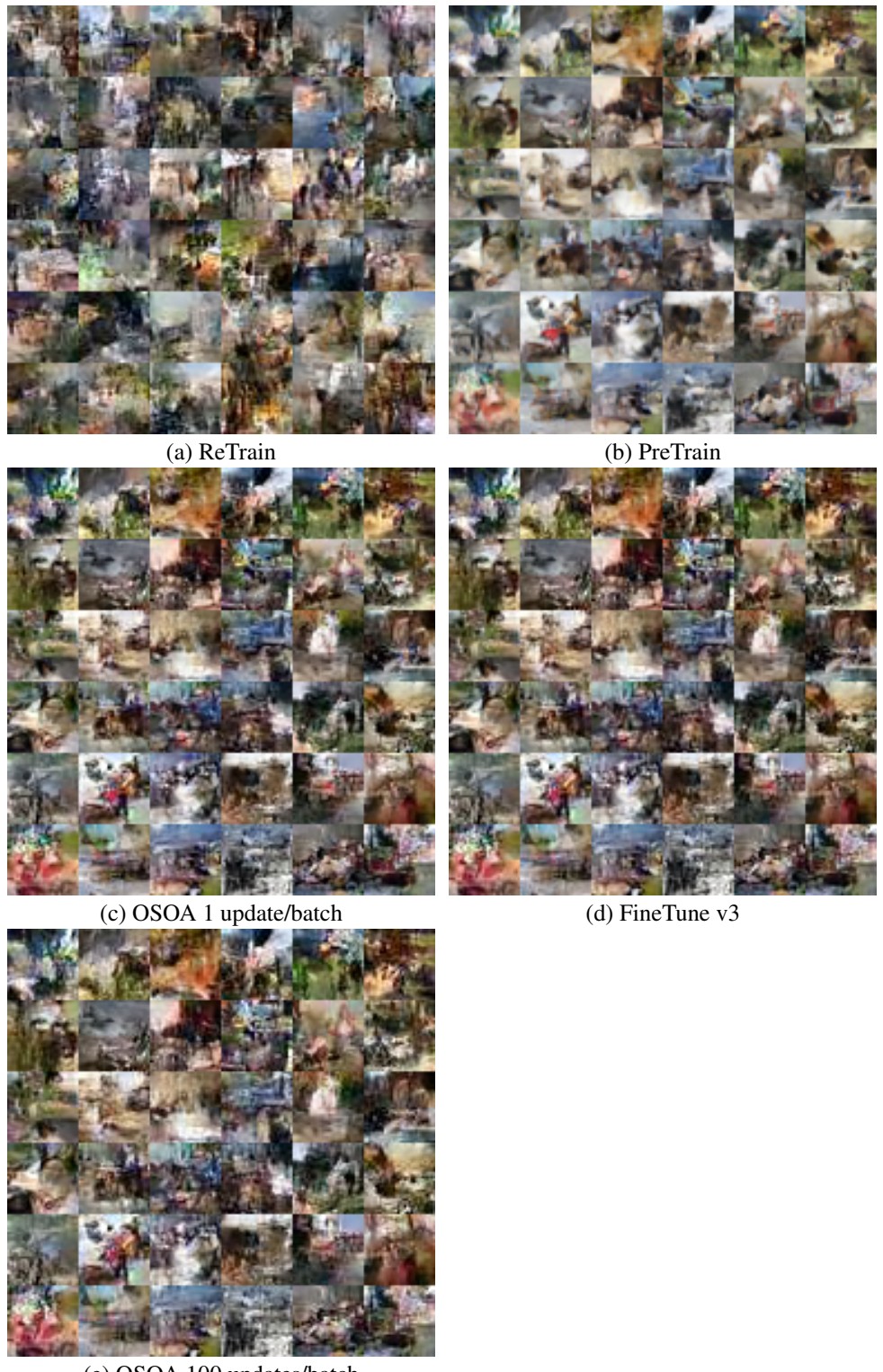

(a) ReTrain

(b) PreTrain

(c) OSOA 1 update/batch

(d) FineTune v3

(e) OSOA 100 updates/batch

Figure 7: 36 images sampled from (a) ReTrain: HiLLoC trained on SET32 from scratch, (b) PreTrain: CIFAR10 pretrained HiLLoC, (d) FineTune v3: fine-tuning CIFAR10 pretrained HiLLoC for 20 epochs on SET32, (c) and (e) OSOA 1 and 100 updates/batch: the final checkpoint of vanilla OSOA with 1 and 100 update steps per batch from CIFAR10 pretrained HiLLoC, respectively.

### B.3 Experiments resources

The infrastructure for experiments of HiLLoC is Intel(R) Xeon(R) CPU @ 2.60GHz×16 CPU with an Nvidia V100 32GB GPU. The CPU used in IDF++ is the same as that in HiLLoC while the GPU is Nvidia P100 16GB GPU.

### B.4 Training details

**HiLLoC pretraining** For the RVAE model (HiLLoC), we adopt the same architecture as the one used in the HiLLoC paper, as specified in Sec. 4.1. We use the amount of free bits 0.1, learning rate 0.002 and batch size 16 for pretraining on CIFAR10 and ImageNet32. For the IAF RVAE model, we adopt the same configuration as RVAE with additionally enabling IAF for the approximate posterior. We use the amount of free bits 0.1, learning rate 0.002 and batch size 32 for pretraining on CIFAR10 and ImageNet32. HiLLoC has 40998823 trainable float32 parameters and IAF RAVE has 49864615 trainable float32 parameters. All of the pretrained models are pretrained for more than 60 hours.

**HiLLoC OSOA** For OSOA in HiLLoC and IAF RVAE, we use the random seed 14865 and learning rate at 0.0002 for all experiments unless otherwise specified. Since there are $2^{17}$ images in total for SET32/64/128, the vanilla OSOA without early stopping is conducted for 512, 2048 and 8192 steps respectively for SET32/64/128.

**IDF++ pretraining** For the IDF++ model, we adopt the same architecture as that in IDF++ paper. For pretraining, we use almost the same hyper-parameter used in IDF [2]. IDF++ has about $56.8M$ trainable parameters.

**IDF++ OSOA** For OSOA in IDF++, we use the random seed 5 and learning rate at 0.0003 for all experiments unless otherwise specified.

### B.5 Performance by changing the hyperparameters

**Error bars with different random seeds** To evaluate the sensitivity of vanilla OSOA with respective to random seeds, we randomly choose 10 different random seeds for CIFAR10 pretrained HiLLoC and 6 random seeds for CIFAR10 pretrained IDF++ on SET32, and show the violin plots in Fig. 8. One can see that OSOA admits performances with plausible consistency of different random seeds.

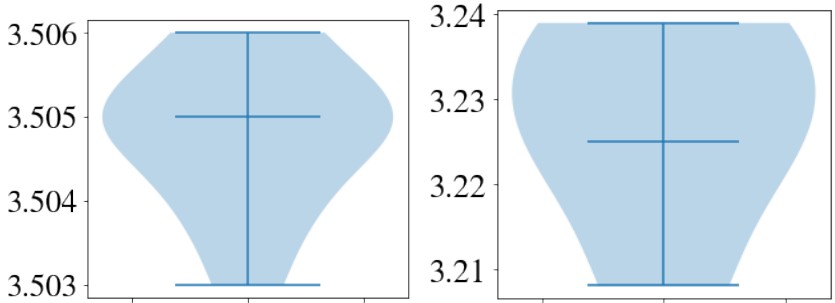

Figure 8: Violin plots of OSOA with different random seeds shown with CIFAR10 pretrained HiLLoC (left) and CIFAR10 pretrained IDF++ (right).

**OSOA with different learning rates** Learning rate is one of the most important factors for fine-tuning and a proper learning rate is usually more desirable. Here we line-searched the interval $[0.0001, 0.012]$ for OSOA with HiLLoC and $[0.0001, 0.001]$ with IDF++, and showcase the learning rate smile curve in Fig. 9. The random seed for HiLLoC is fixed at 14865 and the random seed for IDF++ is fixed at 5. We find for HiLLoC, while the pre-training learning rate is 0.002, a slightly larger learning rate 0.003 can achieve better OSOA performance. For IDF++, a smaller learning rate is preferred, while larger ones lead to the failure of training, which highlights the importance of a proper learning rate choice. Moreover, one can see in Fig. 9 that even with the learning rate difference at the order of magnitude of 0.001, a difference of bpd at the magnitude of 0.1 can be witnessed.

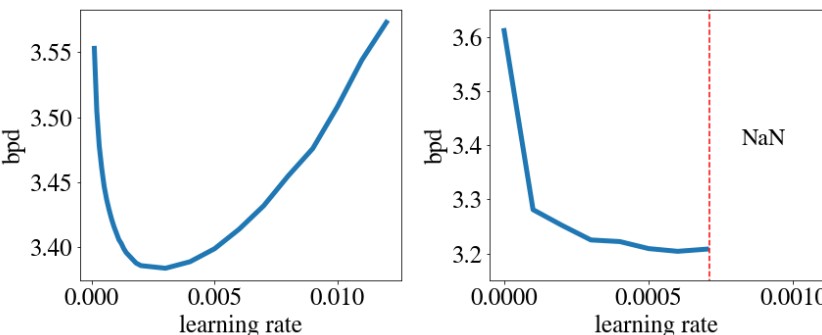

Figure 9: The learning rate smile: bpd of OSOA with HiLLoC (left) and IDF++ (right) of different learning rates on SET32. "NaN" denotes the failure of training with larger learning rates.

### B.6 Real bpd values

Note that the theoretical bpd values reported in this paper are evaluated based on the discretised distributions of models for coding. Depending on the design and implementation of the codec, the real bpd value will be slightly higher than the theoretical one, e.g., an extra cost of less than 32 bits [8] for bits-back ANS, which is negligible for large size datasets [8, 9]. Since the comparison between OSOA and baselines are for the same model and same codec, theoretical bpd results are sufficient for comparison. The codec for HiLLoC is Craystack [8], which is a prototype purpose python implemented codec developed in the work [8] and for IDF++ is the self-implemented AC coder. The real bpd values of HiLLoC and IDF++ on SET32 are shown in Tab 3. One can see that the differences between the theoretical bpd values and the real bpd values for OSOA and FineTune baselines are at the order of magnitude of 0.01 for HiLLoC and the order of magnitude of 0.001 for IDF++. Further, the conclusion of the comparison between the space efficiency of OSOA and FineTune baselines holds.

Table 3: Real bpd values

|  | SET | PRETRAIN | OSOA | FINETUNE V1 | FINETUNE V2 | FINETUNE V3 |
|---|---|---|---|---|---|---|
| HiLLoC (CIFAR10) | 32 | 4.025 | 3.565 | 3.421 (0.144) | 3.364 (0.201) | 3.257 (0.308) |
| IDF++ (CIFAR10) | 32 | 3.614 | 3.227 | 3.092 (0.135) | 3.047 (0.180) | 2.837 (0.390) |