# OpenReview forum: "OSOA: One-Shot Online Adaptation of Deep Generative Models for Lossless Compression"
_NeurIPS.cc/2021/Conference — NeurIPS 2021 Poster_

### Official Review · Reviewer_Tx3u · 2021-07-14

**Rating:** 7
**Confidence:** 4

**Summary:**

The paper considers the task of online adaptation of pre-trained lossless compression models, while simultaneously using the updated model to compress the target dataset. The resulting framework, and new algorithm called OSOA, is based on the key insight that when losslessly compressing many batches of data, one can update the existing model on the data batches on the fly, so that the compression model better captures the target data statistics. Together with a FIFO codec (such as arithmetic coding) or FILO codec with a caching mechanism, the decoder can decode the data batches while replaying the model updates, so that the model update comes with no extra bitrate cost. Experiments demonstrate significant time and space savings of OSOA compared to the baselines of re-training or fine-tuning.


**Limitations And Societal Impact:**

**Limitations**

1. As the authors noted, the fine-tuning baseline requires an extra model update transmission overhead compared to the proposed OSOA method; however with the naive strategy (storing the updates as 32-bit floats, without any entropy-cost-penalty on the updates), this overhead might be over-estimated. The model update overhead from fine-tuning can be significantly reduced with learned entropy coding, e.g., by adapting the technique of Rozendaal et al. (ICLR 2021) to overfit on the entire test batch in a lossless compression setting. A stronger fine-tuning baseline would make a more interesting comparison.
2. The requirement for the decoder to be able to carry out model training (and perform the exact same updates as the encoder) may limit the applicability of the method, e.g., to edge devices.

**Main Review:**

**Originality:**

Although the idea of adapting the model on the data being compressed is not entirely new (e.g., it's been used in traditional codecs like arithmetic coding), the paper successfully applies this idea to learned lossless compression based on deep generative models, which I believe is a novel contribution.



**Clarity:**

The writing can be improved, especially in the background section, where some of the text is sloppily written and might be misleading to a non-expert reader. Examples:
1. lines 77-79: "Another component in a DGM based lossless compression algorithm is the entropy coder, which converts the data symbols into code and the optimal codelength is the distribution" -- it'd be clearer to write "... into code *(bit-strings)* and the optimal *average* codelength is the diisttrubtion of the *ground truth data* distribution".
2. line 82: the statement "AC and ANS perform slightly slower than Huffman coding, they provide a code length closer to the optimal." is imprecise and possibly misleading.
3. The overall problem setup ("maximum likelihood training") can also benefit from more rigor, in order to establish the correct objective functions for the proposed adaptation process. Although not clearly specified, the "distributions" $p^*(x)$ and $p_\theta(x)$ in Eqs (1) and (2) are understood to be probability *density* functions (according to the context, and the subsequent discussion on discretization schemes), and therefore the MLE objectives (1) and (2) don't directly have compression interpretations. This can be fixed by either specifying that that data $x$ is assumed to be discrete, so that $p^*$ and $p_\theta$ are probability *mass* functions (so that (1) and (2) are compression costs), or take the dequantization approach (i.e., adding dequantization noise to the data) and modify (1) or (2) accordingly so that they correspond to upper bounds on the compression cost.


**Quality:**

The claims and evaluation methodology appear mostly sound. As noted above, the clarity of the paper can be improved.
On a related note, I have a clarifying question on the evaluation method: are the theoretical bpd values derived from the same training objectives as the base models (i.e., NELBO for VAE model and NLL for IDF++)?


**Significance:**

The paper tackles a relevant practical problem of deploying a trained compression model on out-of-distribution data. The proposed method broadly applies to generative models for lossless data compression, and should be of interest to most researchers and practitioners working in learned data compression.




**Time Spent Reviewing:**

4

---

> ### Author Response · Authors · 2021-08-10
> **Thanks for the feedback**
>
> The authors sincerely thank the reviewer for the positive feedback and appreciate the reviewer for detailed reading and advice on improving our manuscript. We will revise the paper accordingly and will add more discussion on conventional adaptive compression algorithms for a more comprehensive related work section. Please find following our response to the reviewer’s comments.
>
> -	Thanks for the suggestion to improve the rigor and we will revise line 77-79 according to the suggestion.
> -	Depending on the final page length, we may remove this statement (line 82) or expand it in terms of the computational complexity of these coders and their ability to achieve the distribution entropy with their generated average code length.
> -	We agree with the reviewer that entropy coding is within the scope of discrete distributions only and thank the reviewer for the two possible modification ideas. We will more explicitly highlight the compression objective in the discrete setting to avoid potential confusion.
> -	The theoretical bpd values are evaluated with the models’ default functions for bpd evaluation, i.e., NELBO based for the ResNet VAE model in HiLLoC and NLL based for IDF++. And both models by construction define discrete distributions in the image space. Note that for HiLLoC, a training technique called free-bits is used for the objective function during training but not used for theoretical bpd evaluations.
> -	While recent works show that network compression techniques can contribute to the overall description length reduction, it can add another layer of complexity to stipulate appropriate priors for complicated models and time cost for tuning additional parameter. Therefore we decided to put it as a separate work.
> -	The current version of OSOA, i.e., vanilla OSOA, does depend on gradient evaluation during both encoding and decoding, which can influence the efficiency. It is therefore desirable to develop more efficient OSOA algorithms in future work.

---

> > ### Comment · Reviewer_Tx3u · 2021-08-28
> > **Re: Thanks for the feedback**
> >
> > I thank the authors for the response, which addressed my concerns.

---

### Official Review · Reviewer_9cFX · 2021-07-17

**Rating:** 8
**Confidence:** 4

**Summary:**

The paper introduces One-Shot Online Adaptation (OSOA) for lossless compression by generative models. This method allows a model to adapt to a test-time distribution that is different from the train-time distribution without having to transmit a re-trained model. The basic idea is that one can apply a deterministic learning procedure to the model for each batch in a sequence of batches being compressed. Since compression is done losslessly, the decoder can replicate the exact same training procedure, and thus have access to the same adapted model as was used for compression. The idea is validated in a thorough experimental study.

**Limitations And Societal Impact:**

Some limitations could be better discussed (see above).

**Main Review:**

This paper is based on a really clever idea. Although there are some practical drawbacks (see below), I think this method can really improve compression results significantly. This is already confirmed by the experimental results in this paper, but can probably be pushed further.

The experiments are clean and simple and show a clear benefit of OSOA. Enough details are provided to enable reproducibility.

The paper is fairly well written, although I think it can be made a bit easier to read still. Some details are hard to find, such as whether you count the cost of the model parameters as part of the bpd.

Although I think the basic idea of OSOA is new in the DGM compression literature, I’m quite sure the idea of adaptation / “online learning” is well known in the classical (non DL) compression literature. It would be good to refer to older works that use this idea.

Some downsides of OSOA, that would be good to discuss at more length in the paper
Getting deterministic training to work in practice, across different devices, will be very challenging
Not only encoding time, but also decoding time is greatly increased because it will require training.

Overall I think this is an innovative paper with a solid experimental validation, that is likely to spur further work in the DGM compression literature.


**Time Spent Reviewing:**

3

---

> ### Author Response · Authors · 2021-08-10
> **Thanks for the feedback**
>
> The authors thank the reviewer for the appreciation of our work and we do hope the presentation of our work can help the community push forward the frontier of DGMs based lossless compression. Please find following more details to help further clarification.
>
> -	For the bpd, we convert the model storage size into effective bpd in the last column in Table 2. For the columns of Pretrain, ReTrain and FineTune baselines, the bpd values are for the compressed data only. As a result, when comparing OSOA to baselines, the space cost of baselines should be the baseline data bpd + Model bpd. For OSOA, since the base model storage can be amortised on many target datasets and is regarded as part of the compression software, we do not include the model effective bpd for OSOA.
> -	We agree with the reviewer that such kind of adaptation paradigm can be traced back to conventional non-learning based lossless compression literature. And we will add more relevant literature and discussion into the related work section.
> -	We partially discuss the deterministic training topic in the experimental section and can expand further based on that. Deterministic training is currently not a very well-developed feature in main stream deep learning frameworks but has been drawing more and more attention and developments recently.
> -	Depending on the specific reason causing non-deterministic behaviours and specific implementation to remove the non-determinism, the running time may or may not be affected. Moreover, current version of OSOA, i.e., vanilla OSOA, does depend on gradient evaluation during both encoding and decoding, which can influence the efficiency. It is therefore worthwhile to investigate more sophisticated and efficient algorithms for OSOA, e.g., gradient-free meta learning or reinforcement learning methods.

---

> > ### Comment · Reviewer_9cFX · 2021-08-17
> > **Response**
> >
> > Thanks for the clarifications.

---

### Official Review · Reviewer_d4fb · 2021-07-21

**Rating:** 4
**Confidence:** 4

**Summary:**

This work proposes an adaptive compression scheme that leverages a pre-trained deep generative model (DGM).  Specifically, the authors consider compressing a sequence of many (~thousands or more) samples and propose to periodically update the underlying DGM, so that the DGM gradually becomes a better fit for the dataset at hand.  The main contributions of this work are in its formulation as well as experimental results.

**Limitations And Societal Impact:**

The authors thoroughly discuss potential societal impacts.

**Main Review:**

### Strengths
I find the overall idea of online updating the underlying model interesting.  As authors mention, this could be useful for archival purposes where encoding/decoding speed is not crucial, but the compression ratio is critical.

### Concerns
- **Novelty:** I do not find the proposed formulation to be a sufficient contribution.  Using an entropy model that is adaptive has a long history.  Context-adaptive entropy coders have long been studied [2], and even the online adaptation of the model was explored in [1].  [1] is particularly relevant to this work because it also studies the compression performance of a deep learning model when counting the communication cost of model parameters.  The discovery is that online adaptation of the model leads to a better compression than the two-part scheme that one obtains from a VAE with a prior over model parameters.  Given this, the idea of periodically updating the underlying entropy model during compression seems to be a rather minor extension.
   - I also find that OSOA is compatible with both FIFO and FILO entropy coders (e.g. AC vs. ANS) to be an implementation detail that is irrelevant to the overall paper.  This is because OSOA assumes the existence of a model-specific coder (denoted "inbuilt algorithm of the framework" in the Supplementary Material) and thus is essentially agnostic to the details of it.
  - I am confused by the name of the method as well.  Specifically, what is "one-shot" about the proposed method? The model updates happen periodically for the total of $T$ times over the course of encoding/decoding.

- **Experiments:** While the experimental results are sensible, they do not seem particularly informative.  It seems that they were designed primarily to study the effects of various hyperparameters.  More specifically:
  - Figure 2 (left) does not provide much information, since the ReTrain model is trained from scratch, whereas OSOA benefits from the pre-trained checkpoint.   A similar observation could be made about Figure 2 (middle, right) -- the longer you run OSOA, the more the model is essentially trained on the compressed data.  Thus, it's expected that the gap between OSOA and FineTune would gradually decrease.
  - The effect of `updates/batch` presented in Sec 4.2 shows that updating the model parameters more leads to better bpd.  This result is completely expected, as DGMs are generally trained for many (10s ~ 100s) epochs over the dataset.

- **Writing quality:** The overall writing quality of the paper could be improved.  Aside from confusing/unnatural sentences, the paper also contains some vague claims.  For example, Sec 4.3 discusses the mismatch between the goal of DGM training and lossless compression.  As an illustrative example, it mentions the cases where a likelihood-based model assigns high likelihood on out-of-distribution samples.   While this seemingly counterintuitive phenomenon is indeed real, I disagree that there is a mismatch in the objectives of DGM training and lossless compression.  A likelihood-based model (e.g. a normalizing flow) is trained to maximize its likelihood on the training dataset, which directly corresponds to its compression performance of the said dataset.  _This is irrespective of how the model performs on another dataset._


#### Reference
[1] Blier, Léonard, and Yann Ollivier. "The description length of deep learning models." Proceedings of the 32nd International Conference on Neural Information Processing Systems. 2018.
[2] https://en.wikipedia.org/wiki/Context-adaptive_binary_arithmetic_coding

**Time Spent Reviewing:**

3

---

> ### Author Response · Authors · 2021-08-10
> **Thanks for the comments and we hope it can clarify the confusions**
>
> The authors thank the reviewer for the comments and we would like to further clarify the confusions with our response.
>
> Novelty: While adaptive algorithms for entropy coding have long history in the lossless compression literature, different scenarios call for different treatments and the underlying idea to achieve the effective adaptation can vary. For example, the idea of adaptive dictionary methods (e.g., LZ77) is to make the dynamic dictionary converge to the ground truth one of the repeated patterns and the idea of adaptive context modelling methods (e.g., CMIX, CABAC [2]) is to let the transition probability to converge to the ground truth dynamic. Instead, our work propose a novel way to achieve effective adaptation via a weaker convergence of the model and strong inductive bias of DGMs. Here the concept of strong & weak convergence is parallel to the mathematical concept of convergence in strong & weak topology. The model that can generate sample as of target data and assign high probability to samples in the target dataset (e.g., the model obtained by ReTrain) can be regarded as of comparably stronger convergence but computationally more expensive to get. The models generated during OSOA, gradually assigning higher probabilities on the target dataset but not admitting samples as of the target data, is related to comparably weaker convergence but computationally less expensive to get.
>
> We thank the reviewer for providing the reference [1] to further our knowledge and we will include associated discussions in the related work. Although reference [1] and our work both consider the cost of model storage for data lossless compression in the learning based compression context, reference [1] and our work actually provide two different solutions to the same problem. Reference [1] formulates the problem in a supervised learning paradigm, where both the sender and receiver need the base model and an auxiliary dataset for the label transmission, whereas our work formulates in the unsupervised learning paradigm and only the base model is needed. Here we use an illustrative example to highlight the difference. Let’s say Alice wants to transfer the dataset A to Bob. While both [1] and our work require Alice and Bob to agree on a model architecture and both have the parameter value for the model, [1] additionally require an auxiliary dataset B to be stored on both sides of Alice and Bob. The dataset A should ideally have high mutual information with dataset B, otherwise the presence of dataset B will not help too much on reducing the bitrate of dataset A. Therefore, it puts stronger constraints on the family of dataset A that can be effective compressed with a certain overhead dataset B.
>
> -	While the deep generative model and the model-agnostic codec in the codec module allow for plug-and-play, they are not compatible with the dynamic system module if the model-agnostic codec is of the FILO style. Therefore, the Algorithm 2, as an intermediate component to couple the adaptation and the model-agnostic coding, is a necessary and important component in the OSOA framework.
> -	The name one-shot follows the usual usage of one-shot learning, i.e., each sample is only seen once and just one epoch of training is conducted.
>
> Experiments: The experimental section is designed to validate the proposed framework by comparing to benchmarks and investigate it at a fine-grid level to understand its behaviour. And we do not think the fact that OSOA behaves ideally under non-ideal conditions uninformative.
> -	Even a model is pre-trained on dataset A, there is no theoretical induction that how it can perform on (or benefit) dataset B, especially in the context of deep generative models. Figure 1 Left is not only to experimentally verify the benefit, but also to highlight the huge amount of additional time needed by ReTrain to catch up. Furthermore, together with Figure 3, Figure 2 Left shows that OSOA is supported by a much weaker sense of convergence. In other words, although the loss function decreases on the target dataset, the distributions do not converge to the ground truth model that can generate samples from the target dataset.
> -	Note at each time step t during OSOA, the model to compress the batch_t is only optimised by batches before and excluding batch_t. However, in FineTune, only the final fine-tuned model is used to compress the full target dataset and all data points contribute to the optimisation of the final model. Although different batches do not admit direct dependency, the fact that previous batches do help the compression improvements of following batches indicates the connection between batches at the local feature level and validates its usefulness for data compression.
> -	For experiments of multiple steps per batch, we are showing optimisation with batch t for more steps can benefit batch t+1. It can be expected that optimisation with batch t for more steps can benefit the compression of batch t, which is not the case here. Note that we consecutively conduct multiple optimisation steps with each batch in a single epoch. With knowledge of optimisation, this algorithm is very different from conducting single step per batch for multiple epochs.
>
> Writing quality: We thank the reviewer for advice on improving writing. In Section 4.3, we are discussing that data generation purpose modelling and compression purpose modelling may not be strictly equivalent. We are not discussing the mismatch between objective function in DGM training and lossless compression. Here we give an illustrative example to explain our point. Assume one has a dataset of dogs and a pre-trained model of cats. If one would like to generate samples of dogs, one has no other choice but to training a model on the dataset of dogs. But if one would like a model to compress the dataset of dogs, one can actually use the model of cats to get a decent compression ratio (message from recent works). Further, one can actually use the datasets of dogs to adapt the model of cats and get a much better compression ratio – even though after adaptation it’s still a model of cats (message from our work and cf. Figure 3).

---

> > ### Comment · Reviewer_d4fb · 2021-08-27
> > **Thank you for the clarifications.**
> >
> > Thank you, I appreciate the clarifications provided by the authors.

---

### Official Review · Reviewer_s1Vi · 2021-07-21

**Rating:** 4
**Confidence:** 4

**Summary:**

This paper presents a method for adapting a learned compression model to improve the lossless compression rate over a set of data to be transmitted, e.g. a set of images. The method assumes a generative model has been optimized over a large dataset, and several models are explored empirically including HiLLoC and IDF++. The model represents a distribution over data items that can be used as the entropy model for a lossless compression algorithm (Huffman coding, arithmetic coding, ANS, etc.). The core idea in OSOA is to divide the data to be compressed into separate batches and to adapt the pre-trained model on each batch before coding the next batch.

The authors show that this approach reduces total bit rate, outperforms training and transferring a separate model for each batch (at least in several cases), that the method can be adapted to FIFO and FILO algorithms (e.g. range coding vs ANS), and that there is a trade-off between runtime and compression rate based on early stopping of the update algorithm.

**Limitations And Societal Impact:**

yes

I thought the authors did a very good job pointing out potential societal impacts "hidden" in the generative models. Although the models are used for compression, they could be extracted and sampled, which could leak information from the training set (see Figure 3). This is a great point and warning on the part of the authors.

**Main Review:**

Originality = 2/5. While I haven't see this kind of online adaptation before for deep generative models, it does follow the general pattern of "backward adaptation", which is very common in image and video coding.

Quality = 4/5. The authors do support their claims with various empirical evaluations including comparison multiple methods for fine-tuning a model. What's not clear is that benefits of OSOA hold up for larger images (or other large data items). This can be seen in Table 2, where OSOA provides a huge rate savings for small images (32x32) since the effective bpd of the model is large compared to the 1024 pixels in each image. For 128x128 images, however, the bpd of the model is smaller (now there are 16,384 pixels) so the most aggressive finetuning approach maintains a small rate advantage of OSOA. OSOA may have a runtime advantage in this case, but it's a less clear net advantage.

Clarity = 4/5. The paper was fairly easy to follow but could use another round of editing. Perhaps restructuring how the FIFO and FILO variations are presented could help, e.g. present just the simpler FIFO case first so that readers understand the overall structure, and then discuss what changes for FILO.

Significance = 2/5. Although I think the problem setting is interesting, the relatively low novelty (at least at a conceptual level) hinders overall significance.

**Time Spent Reviewing:**

2

---

> ### Author Response · Authors · 2021-08-10
> **Backward adaptation is a high level principle and we propose a novel way to instantiate it**
>
> The authors thank the reviewer for the comments and the suggestion to present the FIFO and FILO variants. We hope the response can help make further clarification.
>
> Originality and significance: At the high level, all online adaptive compression algorithms should follow the paradigm of backward adaptation, in order to recover the model parameter, used to encode $x_t$, with only $x_1, ..., x_{t-1}$ at the decoding stage. Nevertheless, practices instantiating the high level idea for different underlying static models need to accommodate the specifics and require different insights. Adaptive dictionary methods, e.g., LZ77, dynamically maintain a dictionary of patterns to gradually recover the ground truth dictionary containing the repeated patterns in the sequence. Adaptive context modelling methods, e.g., CMIX and PAQ, allow dynamic adjustments of the context model to converge to the ground truth transition probability of the sequence. In this work, we introduce a novel way to realise backward adaptation, which is to leverage the convergence in a weaker sense with strong inductive bias of deep generative models. Specifically, one does not need to adapt the model to one that can both assign high probability to the target data and generate samples as the target data (mathematically convergence in a stronger sense, and practically more expensive), but only needs to adjust the pre-trained model locally for higher probabilities of data in the target dataset (mathematically convergence in a weaker sense, and practically less expensive), cf. results and discussion in Section 4. Moreover, popular context modelling methods for image and video compression build models for the transition probability between pixels or frames, where strong dependencies are assumed. Instead, our work focuses on joint distribution of elements in in data variable and does not necessarily require dependency in the data sequence, of which the setting could benefit from a separate set of principles with our work as a preliminary exploration.
>
> Quality: Note that each dataset in SET32/64/128 contains $2^{17}$ = 131072 images respectively. The storage cost of model in FineTune is amortised by the number of images. While larger images sizes and larger image numbers will potentially require models of larger scales, here we fix the model size and show its impact on different numbers of images for SET128. By fixing the order of images as the one in OSOA, if we only compress the first $2^{16}$, $2^{15}$ and $2^{14}$ images (128*128 size), the FineTune V2 admits bpd, in the code bpd + model bpd format, 2.564 + 0.408, 2.611 + 0.816 and 2.658 + 1.632. However, OSOA admits bpd, in the code bpd format since no model storage required, 2.734, 2.788 and 2.842, which are 0.238, 0.639 and 1.448 less than FineTune V2 respectively. To conclude, OSOA remains a clear net advantage than FineTune with the same time budget up to a reasonable large sample size.

---

### Decision · Program_Chairs · 2021-09-27

**Decision:**

Accept (Poster)

**Comment:**

The paper proposes to use a deep generative model optimized over a large dataset as the entropy model for a lossless compression algorithm. The main idea is to divide the data to be compressed into separate batches and to adapt the pre-trained model on each batch before coding the next batch.
There are two reviewers opposing the acceptance of the paper and two reviewers who are for accepting it. The main drawbacks pointed out are:
- Potentially limited originality because “backward adaptation” is commonly used in image and video coding.
- Writing quality could be significantly improved.
- Some experiments seem to be less informative.
- Older (non-DL) literature is missing.

All reviewers appreciated the quality and clarity of the paper. Two reviewers especially valued the idea of adapting the model on data being compressed with deep generative models. The rebuttal was positively received by the reviewers. In my assessment, even though the originality of the paper is put in question, the paper is an interesting contribution to the field of data compression with deep generative modeling. Therefore, I tend to accept the paper.